# Ectopic Expression of *CrPIP2;3*, a Plasma Membrane Intrinsic Protein Gene from the Halophyte *Canavalia rosea*, Enhances Drought and Salt-Alkali Stress Tolerance in Arabidopsis

**DOI:** 10.3390/ijms22020565

**Published:** 2021-01-08

**Authors:** Jiexuan Zheng, Ruoyi Lin, Lin Pu, Zhengfeng Wang, Qiming Mei, Mei Zhang, Shuguang Jian

**Affiliations:** 1Guangdong Provincial Key Laboratory of Applied Botany & Key Laboratory of South China Agricultural Plant Molecular Analysis and Genetic Improvement, South China Botanical Garden, Chinese Academy of Sciences, Guangzhou 510650, China; zhengjiexuan16@mails.ucas.ac.cn (J.Z.); linry@scbg.ac.cn (R.L.); pulin@scbg.ac.cn (L.P.); 2University of the Chinese Academy of Sciences, Beijing 100039, China; 3Key Laboratory of Vegetation Restoration and Management of Degraded Ecosystems, Center for Plant Ecology, Core Botanical Gardens, Chinese Academy of Sciences, Guangzhou 510650, China; wzf@scbg.ac.cn (Z.W.); mei4597@scbg.ac.cn (Q.M.); 4Southern Marine Science and Engineering Guangdong Laboratory (Guangzhou), Guangzhou 511458, China; 5Center of Economic Botany, Core Botanical Gardens, Chinese Academy of Sciences, Guangzhou 510650, China; 6CAS Engineering Laboratory for Vegetation Ecosystem Restoration on Islands and Coastal Zones, South China Botanical Garden, Chinese Academy of Sciences, Guangzhou 510650, China

**Keywords:** plasma membrane intrinsic proteins, salt-alkali, drought, *Canavalia rosea*

## Abstract

Aquaporins are channel proteins that facilitate the transmembrane transport of water and other small neutral molecules, thereby playing vital roles in maintaining water and nutrition homeostasis in the life activities of all organisms. *Canavalia rosea*, a seashore and mangrove-accompanied halophyte with strong adaptability to adversity in tropical and subtropical regions, is a good model for studying the molecular mechanisms underlying extreme saline-alkaline and drought stress tolerance in leguminous plants. In this study, a PIP2 gene (*CrPIP2;3*) was cloned from *C. rosea*, and its expression patterns and physiological roles in yeast and *Arabidopsis thaliana* heterologous expression systems under high salt-alkali and high osmotic stress conditions were examined. The expression of *CrPIP2;3* at the transcriptional level in *C. rosea* was affected by high salinity and alkali, high osmotic stress, and abscisic acid treatment. In yeast, the expression of *CrPIP2;3* enhanced salt/osmotic and oxidative sensitivity under high salt/osmotic and H_2_O_2_ stress. The overexpression of *CrPIP2;3* in *A. thaliana* could enhance the survival and recovery of transgenic plants under drought stress, and the seed germination and seedling growth of the *CrPIP2;3 OX* (over-expression) lines showed slightly stronger tolerance to high salt/alkali than the wild-type. The transgenic plants also showed a higher response level to high-salinity and dehydration than the wild-type, mostly based on the up-regulated expression of salt/dehydration marker genes in *A. thaliana* plants. The reactive oxygen species (ROS) staining results indicated that the transgenic lines did not possess stronger ROS scavenging ability and stress tolerance than the wild-type under multiple stresses. The results confirmed that *CrPIP2;3* is involved in the response of *C. rosea* to salt and drought, and primarily acts by mediating water homeostasis rather than by acting as an ROS transporter, thereby influencing physiological processes under various abiotic stresses in plants.

## 1. Introduction

*Canavalia rosea* (Sw.) DC., also known as bay bean, is mainly distributed in coastal and island sandy soils in tropical and subtropical areas. It is a perennial halophyte belonging to the Fabaceae family that has adapted to extreme environments with high temperature and strong light in coastal areas and exhibits significant salt-alkali tolerance and drought resistance [1,2]. In particular, *C. rosea* shows great ecological adaptability on tropical coral reefs, which makes it a candidate greening species in restructuring the ecological functions of marine and island ecosystems. As a sea-dispersed legume with a high level of nutrient utilization efficiency and good nitrogen-fixing capacity, *C. rosea* constitutes a superior wild plant resource and plays important roles in wind resistance, sand fixation, landscape greening, and ecological restoration in the vegetated areas on coral islands and in coastal zones [1]. In addition, due to its excellent salt-alkali and desiccation tolerance to extreme environments, *C. rosea* offers a superior genetic resource pool to identify abiotic stress tolerance genes, especially salt/drought resistance genes involved in osmotic and water deficit tolerance, for the further abiotic-stress-related genetic improvement of leguminous plants or other glycophytic crops.

Aquaporins (AQPs) are highly conserved integral membrane channel proteins, also known as major intrinsic proteins (MIPs), that are mainly involved in water movement across membranes in cells [3,4]. AQPs show highly conserved structures whereby six membrane-spanning alpha helices are linked by five loops with their N- and C-termini into the cytosol. Two of these loops contain highly conserved asparagine–proline–alanine (NPA) motifs, which are of importance in the formation of water-selective channels [5,6]. AQPs are widely distributed in all living organisms, from prokaryotes to plants and animals, and their ability to transport water and other neutral small molecules has been confirmed by a series of experiments in vivo or in vitro [7]. In plants, many AQPs have been reported to play essential roles in regulating water potential and transport in vascular plants [8,9]. Genome sequencing has indicated that plant AQPs constitute a large gene family, basically containing tens of members that can be classified into five subfamilies based mainly on their subcellular localization [6]. These subfamilies include plasma membrane intrinsic proteins (PIPs), tonoplast intrinsic proteins (TIPs), nodulin-like plasma membrane intrinsic proteins (NIPs), small intrinsic proteins (SIPs), and X intrinsic proteins (XIPs). Among them, plant PIPs are the largest group of plant aquaporins and are considered the main water transport pathway across plasma membranes in root and leaf tissues, thus playing important roles in plant–water relations [10].

PIPs can be further divided into two subclasses, PIP1s and PIP2s, based on the length of the N- and C-termini of the PIPs [11]. In plants, it was shown that PIP2s function as water channels when expressed in *Xenopus oocytes*, whereas PIP1s generally have much lower or no water channel activity [12]. Several reports indicated that PIP1 and PIP2 aquaporins may interact to increase water permeability [10,11]. The water permeability of PIP1s requires the co-expression of PIP2s to form hetero-tetramers [13]. PIPs play diverse and important roles in many cellular processes, such as water deficits [14,15], pathogen invasion [16,17], metalloid or metal transport [18,19,20], gaseous and signal molecule transduction [21,22], or other small molecules such as glycerol [23] and urea [24]. Plant *PIP* gene expression is differentially regulated in various tissues and is also altered under different physiological and environmental stresses, including abiotic stresses and plant hormones [25,26], especially in the case of drought and salinity tolerance [7,8]. Many researchers have revealed the significant role of plant PIPs in acquiring abiotic stress tolerance from different perspectives, including transcriptional analysis [14], plant biotechnology [27,28], subcellular localization patterns [29], protein modifications [30], and protein interactions [29,31,32].

In recent years, there have been numerous reports that the overexpression of *AQPs* in transgenic plants increases resistance to abiotic stresses [27,28], therefore proposing the genetic manipulation of plant AQPs as a new perspective for the improvement of the physiological responses of crops to abiotic stresses. Notably, the functional identification of non-model plant AQPs, especially of plant species with specialized habitats, has attracted considerable attention recently. The four-wing saltbush *Atriplex canescens* is a temperate halophyte with excellent saline-alkaline tolerance, and its plasma membrane intrinsic protein gene, *AcPIP2*, improved salt and alkali tolerance, leading to increased sensitivity to drought stress, when heterologously overexpressed in *Arabidopsis thaliana* [33]. Another *A. canescens* nodulin-like plasma membrane intrinsic protein gene, *AcNIP5;1*, showed a totally opposite phenotype when expressed in *A*. *thaliana* [34]. *Sesuvium portulacastrum* is a perennial halophyte that typically grows in coastal and inland sandy soils and exhibits excellent tolerance to salt. *SpAQP1* (a PIP2 member) enhanced the salt tolerance of yeast strains and tobacco plants and promoted seed germination and root growth under salt stress in transgenic plants [35]. *Selaginella moellendorffii* is a desiccation-tolerant plant, and a total of 19 *AQP*s were detected in the *S. moellendorffii*-expressed sequence tag (EST) database, while only three conserved PIPs were detected [36]. *SmPIP1;1* and *SmPIP2;1* exhibited different water-channel activities, and their co-expression showed a synergistic effect on the water membrane in an oocyte system, while no synergistic effect was observed in yeast [37]. *Stipa purpurea* showed great adaptability to stresses such as drought and a changing environment in the meadows of the Tibetan Plateau [38]. The transcription of *SpPIP1* increased significantly under drought, and the ectopic expression of *SpPIP1* in *A. thaliana* elevated the drought tolerance of the plants [38]. Jojoba (*Simmondsia chinensis*) is a typical desert plant with strong tolerance to drought, salinity, and nutrient-poor soils, and *ScPIP1* could improve salt and drought resistance when expressed in *A*. *thaliana* [39]. *Thellungiella salsuginea* displays strong resistance to high salinity, drought, and chilling stress. The overexpression of *TsPIP1;1* in rice could also enhance the salt tolerance of rice by maintaining the osmotic potential and promoting photosynthesis [40]. Both *NnPIP1-2* and *NnPIP2-1*, from the hydrophyte salt-tolerant lotus *Nelumbo nucifera*, could enhance the abiotic stress tolerance of transgenic plants when overexpressed in *A. thaliana* [41]. These findings indicate that *PIPs* from plants growing in specialized habitats can be excavated as potential target genes for enhancing plant resistance to abiotic stresses though genetic engineering.

To further understand the mechanism of salinity tolerance in *C. rosea*, an entire cDNA library was constructed from *C. rosea* seedlings for yeast functional screening. A series of ESTs were previously characterized, including *CrPIP2;3*. In this study, this AQP gene *CrPIP2;3* was further functionally characterized based on an over-expression assay in yeast and *A. thaliana.* Our results suggested that *CrPIP2;3* was significantly induced by high osmotic stress and could enhance the salt–alkali and drought tolerance of transgenic plants by facilitating water transport and increasing the drought tolerance of the plants.

## 2. Results

### 2.1. Identification and Bioinformatics Analysis of CrPIP2;3

The full-length cDNA of *CrPIP2;3* was isolated from the cDNA library constructed using *C. rosea* seedlings. The length of the *CrPIP2;3* cDNA is 1254 bp, with a 51-bp 5′ untranslated region (UTR) and a 333-bp 3′ UTR (GenBank Accession No.: MT787666). The open reading frame (ORF) is 870 bp, encoding a protein with a predicted molecular weight of 31.07 kDa and a theoretical isoelectric point of 7.66. CrPIP2;3 was characterized as an aquaporin with an MIP (major intrinsic protein) domain (33 aa to 269 aa, PF00230) (Figure 1A). The instability index (II) of CrPIP2;3 is 36.11, and the grand average of hydropathicity (GRAVY) is 0.383, while the aliphatic index (AI) showed a relatively high-level value of 99.31, which indicated that *CrPIP2;3* encoded a stable hydrophilic protein. The three-dimensional structure predication by Phyre2 showed that CrPIP2;3 has six transmembrane α-helices (TM1 to TM6) (Figure 1B). Sequence analyses also indicated that CrPIP2;3 contained two conserved Asn-Pro-Ala (NPA) motifs (LB and LE loops) and a putative MIP signal sequence (SGxHxNPAVT) in loop B, with several relatively conserved amino acid residues, including aromatic/arginine (ar/R) selectivity filters (H2, H5, LE1, and LE2) and Froger’s positions (FPs) (P1, P2, P3, P4, and P5) (Figure 1A), which are associated with the substrate selectivity of AQPs [42,43]. Phylogenetic analysis of CrPIP2;3 with other plant species PIPs (including AtPIPs, CaPIPs, and GmPIPs) [44,45,46] showed that CrPIP2;3 is closely related to CaPIP2;2 or GmPIP2;7 and GmPIP2;8, which could be classified into a PIP2 subgroup (Figure 1C). We also predicted the localization patterns of CrPIP2;3 with different online programs, and the prediction results from WoLF PSORT showed that CrPIP2;3 localization at the plasma membrane and Golgi apparatus scored 13 and 1, respectively, while Plant-Ploc predicted CrPIP2;3 to be mainly distributed in the cell membrane. In general, CrPIP2;3 was predicted as a plasma-membrane-localized protein.

Analysis of the upstream 2000-bp sequence from the transcription start site (TSS) of *CrPIP2;3* (promoter region) using PlantCARE indicated that the putative *cis*-acting regulatory elements were classified into three types: the transcription factor-binding site involved in drought inducibility, the hormone-responsive element, and the light-responsive element (Appendix A). The sequence analysis of the *CrPIP2;3* promoter and the putative *cis*-acting element location and function was showed in Appendix A. Generally, the basic promoter elements, TATA-Box and CAAT-Box, were located at −11 and −132 bp, respectively (Figure 2). The *CrPIP2;3* promoter contained three drought-related elements, namely, MYB, MYC, and CCAAT-box, and it harbored CGTCA or TGACG (response to MeJA), the WUN motif (wounding inducible), and several light-responsive elements (AT1-motif, Box-4, GATA motif, I-box, TCCC-motif, and TCT-motif) (Figure 2). This result indicated that *CrPIP2;3* was related to a drought-resistant pathway, with other possible roles in light or biotic stress responses.

### 2.2. Expression Patterns of CrPIP2;3 in C. rosea

To investigate and predict more detailed results regarding the stress responses of *CrPIP2;3*, qRT-PCR was performed to explore the expression patterns under stress or ABA treatments and normal growth conditions using the total RNA extracted from the various tissues. We checked the expression of *CrPIP2;3* in the whole *C. rosea* plant. The result indicated that *CrPIP2;3* was expressed extensively in all tested tissues/organs of *C. rosea* and showed a relatively high transcriptional level with respect to the reference gene *CrEF-α* (Appendix A). In Figure 3, our results revealed that under high-salt (600 mM NaCl) treatment, the expression of *CrPIP2;3* decreased immediately (2 and 12 h) in all three tested tissues when compared with CK samples (the beginnings of each treatment) (Figure 3A). The expression level of *CrPIP2;3* in *C. rosea* roots gradually recovered with the increase in high-salt stress duration (24 and 48 h), while in the vine and leaf, the expression level of *CrPIP2;3* remained obviously decreased until 48 h. Under alkali toxicity (150 mM NaHCO_3_), the expression level of *CrPIP2;3* exhibited correspondingly low levels, with no major changes observed (Figure 3B). The high osmotic stress, achieved by immersing the roots in 300 mM mannitol, significantly induced the expression of *CrPIP2;3* in the root and vine, while the expression of *CrPIP2;3* in the leaf was inhibited (Figure 3C). The ABA treatment also affected *CrPIP2;3* expression (Figure 3D). In brief, in contrast to the other stresses, spraying ABA first repressed the expression of *CrPIP2;3* immediately in the entire *C. rosea* seedlings, following which the expression level recovered slowly both in the root and in the leaf, while in the vine, the expression of *CrPIP2;3* was obviously elevated following ABA treatment at 48 h. We also performed qRT-PCR to assess the expression differences of *CrPIP2;3* in different *C. rosea* organs. Our results showed that the expression level of *CrPIP2;3* in the various parts of *C. rosea* was obviously different (Appendix A). *CrPIP2;3* expression showed differentially regulated patterns in the different tissues of *C. rosea*, which indicated a complex response mechanism and implicated *CrPIP2;3* in the response of *C. rosea* to these abiotic stresses.

### 2.3. Heterologous Expression of CrPIP2;3 Confers Osmotic Stress and H_2_O_2_ Sensitivity in Yeast

The functional identification of *CrPIP2;3* was first performed with a yeast expression system. The cell culture was adjusted to an OD600 value of 1 and then gradient-diluted (to 1:10, 1:100, and 1:1000). Two microliters of yeast solutions were spotted onto SDG agar plates with or without NaCl, sorbitol, or H_2_O_2_. As the results showed, W303 transformed with either *CrPIP2;3* or pYES2 grew normally and did not show growth differences on the SDG control plate. However, with increased NaCl, W303 transformed with *CrPIP2;3* showed an obvious growth lag compared with the yeast containing the pYES2 control (Figure 4A). Similarly, the yeast strain with the pYES2 control also displayed better growth performance on the hyperosmotic plate (plus PEG8000 or sorbitol), which indicated that *CrPIP2;3* could accelerate water loss when the yeast cells survived under water deprivation (Figure 4B,C). We also assessed the H_2_O_2_ transport activity with the yeast expression system. *CrPIP2;3* increased the H_2_O_2_ sensitivity of the yeast when grown on SDG medium containing different concentrations of H_2_O_2_, while both the BY4741 and H_2_O_2_ sensitive mutant strain *skn7Δ* yeast cells showed similar growth performance on the SDG control plate (Figure 4E,F).

### 2.4. CrPIP2 Exhibits Apparent Tolerance to Drought Stress in Transgenic A. thaliana Plants

To further evaluate the effects of *CrPIP2;3*, *A. thaliana* transgenic plants ectopically expressing *CrPIP2;3* under the control of the 35S promoter were generated. After confirmation with genomic PCR (Figure 5A) and qRT-PCR (Figure 5B), three homozygous T3 line plants (*OX 3#*, *OX 6#*, and *OX 10#*) were selected and applied in the following tests. Briefly, about 30 sterilized seeds were spotted on the MS medium with the addition of salt, saline-alkaline, or mannitol stress, following which the germination rate was calculated. As indicated in Figure 5C, a difference in the germination of the WT and *CrPIP2;3 OX* lines could be observed, and the germination of the *CrPIP2;3 OX* lines was higher than that of the WT when the seeds were grown in MS medium containing 200 mM NaCl or 5, 7.5, and 10 mM NaHCO_3_, and the corresponding statistical analysis of the seed germination rates also showed consistent results with the germination status, with better salt and alkali tolerance (Figure 5D,E). There was little difference in the seed germination rates between the WT and transgenic plants under osmotic stress (200, 300, and 400 mM mannitol) (Figure 5C). Although the *CrPIP2;3 OX* lines produced a larger radicle and hypocotyl than WT under 400 mM mannitol stress, the statistical analysis of the seed germination rates did not indicate obvious differences between WT and the three *CrPIP2;3 OX* lines (Figure 5F).

Uniformly growing seedlings (WT and *CrPIP2;3 OX* lines) cultured on MS medium for 7 d were transplanted to MS plates with different stresses to evaluate their tolerance at the seedling growth stage. Basically, the main root length of the WT and *CrPIP2;3 OX* lines was suppressed with the increase in the concentration of NaCl, NaHCO_3_, or mannitol. Although there were no clear differences between the WT and *CrPIP2;3 OX* lines under 150 or 200 mM NaCl, about half of the WT seedlings died under 200 mM NaCl (Figure 6A,B). In the presence of NaCl and NaHCO_3_ (salt-alkali stress), the length of the WT root was slightly shorter than that of the *CrPIP2;3 OX* lines (Figure 6A,C). In accordance with the results at germination, the high osmotic stress (200, 300, and 400 mM mannitol) did not cause obvious differences between the WT and three *CrPIP2;3 OX* lines (Figure 6A,D). These results indicated that the effects of *CrPIP2;3* under the different stresses varied, and the overexpression of *CrPIP2;3* enhanced the salt/alkali tolerance but did not respond greatly to osmotic stress at the germination stage and seedling stage.

Seeds of the WT and *CrPIP2;3 OX* lines were grown under well-watered conditions for 30 d, and before salt, drought, and alkali stress treatment, the growth status of the adult plants (WT and three *CrPIP2;3 OX* lines) was relatively consistent. There was no difference in the tolerance of the adult plants between the WT and transgenic lines (*OX 3#*, *OX 6#*, and *OX 10#*) under salt (200 mM NaCl) or salt–alkali (100 mM NaHCO_3_, pH 8.2) stresses (results not shown). Apparently, *CrPIP2;3* strongly improved the drought tolerance during the growth of the adult plants (Figure 7A). After 13 d of withholding water, the plants were wilted to different degrees in both the WT and three *CrPIP2;3 OX* lines. After re-watering and growth for another 7 d, the WT plants did not recover and appeared to exhibit a lethal phenotype, while most of the *CrPIP2;3 OX* plants in all three lines recovered and remained alive, with an obviously higher survival rate than that of the WT (Figure 7B). This result indicates that *CrPIP2;3* overexpression increased plant resistance to drought.

### 2.5. The Overexpression of CrPIP2;3 Affects the Expression of Genes Related to the ABA Signaling Pathway and Osmotic Stress in A. thaliana

To evaluate the potential impact of *CrPIP2;3* overexpression in stress signal pathways, the expression of marker genes involved in certain stress responses in *A. thaliana* was analyzed by qRT-PCR (Figure 8). *HAI2* (At1g07430) encodes a highly ABA-induced protein phosphatase 2C member, which is responsible for the ABA-activated signaling pathway; the expression of *RD29B* (At5g52300) is induced in response to water deprivation, and this response is mediated by ABA. Under salt stress treatment (200 mM NaCl), both *HAI2* and *RD29B* in the *CrPIP2;3 OX* plants showed obviously elevated expression patterns (Figure 8A), and the expression of *RD29B* was enhanced in the *CrPIP2;3 OX* plants even under normal growth conditions (Figure 8B). Salt-alkali (50 mM NaHCO_3_ at pH 8.2) and high osmotic (400 mM mannitol) stress both induced a higher expression of *RD29B* in *CrPIP2;3 OX* plants than in WT plants (Figure 8B), which also indicated that in the *CrPIP2;3 OX* plants, the high expression of *CrPIP2;3* might affect the ABA response signal pathway, and thereby improve the adaptability to water-deficit stress. Both *RD26* (At4g27410) and *ANAC019* (At1g52890) encode NAC transcription factors, which are involved in ABA-mediated dehydration responses. Both genes were significantly enhanced in *CrPIP2;3 OX* plants under salt, high osmotic, and alkali stress treatments (Figure 8C,D), which also implied that *CrPIP2;3* plays a regulatory role in the ABA-associated water imbalance response signaling pathway.

Accordingly, several *A. thaliana* antioxidant-enzyme genes, including *CAT1* (*At1g20630*), *CSD1* (*At1g08830*), *APX1* (*At1g07890*), and *FSD1* (*At4g25100*), were also evaluated by qRT-PCR, and the results indicated that the expression of these genes was not affected by *CrPIP2;3* overexpression, even under salt, high osmotic, and alkali stress challenges (Appendix A). We also assessed the ROS accumulation status in the WT and three *CrPIP2;3 OX* seedlings under salt, high osmotic, and alkali stress challenges by NBT and DAB staining. As we identified CrPIP2;3 as a H_2_O_2_ transporter in yeast, we hypothesized that this protein might be involved in ROS signal transduction in plants. However, unexpectedly, the overexpression of *CrPIP2;3* in *A. thaliana* did not alleviate the ROS accumulation caused by these above stress challenges, and in the WT and three *CrPIP2;3 OX* lines, the ROS accumulation seemed to be similar, although the ROS level was elevated by these stresses in both the WT and three *CrPIP2;3 OX* lines (Appendix A).

## 3. Discussion

As a mangrove-associated species, *C. rosea* is well adapted to environments with high saline-alkaline levels and is subjected to drought and high osmotic stress in tropical or subtropical coastal areas or islands. It can therefore be considered an “extremophile” due to its unique tissue tolerance adaptations. *C. rosea* also exhibits a remarkable growth rate on coral islands with limited fresh water and nutrient resources, which suggests that this species can be used as a ground cover plant in ecological restoration or reconstruction on tropical islands and reefs to maintain species diversity and landscape characteristics [1,2]. However, the key genes involved in the molecular stress tolerance mechanisms remain unknown. Elucidating the molecular mechanisms of water utilization in this plant could be crucial to explaining its survival under extreme water deprivation. While these genes exhibiting stress resistance might share many common features with homologs from other plant species, there could also be some specific differences considering the extreme growing conditions of this plant. In the present study, we identified and characterized an aquaporin (AQP) gene, *CrPIP2;3*, from *C. rosea* for the first time.

AQPs, belonging to a large protein family, mainly mediate the transmembrane transportation of water and other neutral molecules and are thus involved in plant responses to various water-deficit environmental stresses, including salt, drought, cold, and even desiccation [3,4]. The critical role of *AQP*s in regulating plant stress tolerance has been extensively addressed in multiple plant species over the past decades [6,7]. However, in recent years, there has been an increasing body of research focused on the functional identification of *AQP*s from plants with specialized habitats. Chickpea (*Cicer arietinum*), wild soybean (*Glycine soja*), and common vetch (*Vicia sativa*) are all legumes with better drought or salinity stress tolerance than standard crops. The *AQP* family in these three species has been systematically characterized, providing valuable information for further functional analysis to infer the roles of AQPs in adaptation to diverse environmental conditions [45,47,48]. Psammophytes jojoba (*Simmondsia chinensis*), date palm (*Phoenix dactylifera*), and jujube (*Ziziphus jujuba*) can all tolerate drought, salinity, and nutrient stress, and their *AQP* genes have been intensively studied to better understand their potential for genetic engineering to improve plant stress tolerance [39,49,50]. The halophytes *Atriplex canescens*, *Sesuvium portulacastrum*, and *Thellungiella salsuginea* all exhibit strong salt tolerance, and their *PIP* genes have been found to function in the response to salt stress and could be used in genetic engineering to improve plant growth under abiotic stress [33,35,40]. However, overall, the biological functions of these plant AQP members in stress tolerance are mainly related to their structure and biochemical function in water transport in response to physiological changes in the cellular environment, and this feature appears to have been highly temporally conserved in various species. Among them, plant plasma membrane intrinsic proteins (PIPs) exhibit the most homogeneity and high sequence identity across species and have been widely proved to be involved in abiotic stress resistance [8,10,11]. In this study, the first identified *AQP* gene from the halophyte *C. rosea*, *CrPIP2;3*, was grouped into the PIP2 subfamily and showed high levels of homology to CaPIP2;2 and GmPIP2;5 (more than 80% identity in the amino acid sequence) (Figure 1A,C), which suggests that this protein is important for water channel activity, as observed for other PIP2 subfamily members. Amino acid sequence analysis showed that CrPIP2;3 contains six putative transmembrane α-helices and a conserved MIP signal sequence (SGxHxNPAVT), which is found in all PIP members (Figure 1A,B), and the subcellular localization prediction indicated that CrPIP2;3 was mainly distributed in the plasma membrane, which further defined this protein as a PIP2 transmembrane transport protein with strong water channel activity.

Although numerous studies have shown that plant PIP2s play considerable roles in abiotic stress tolerance, including drought, salt, and even cold, the expression pattern of *AQP*s should provide the most immediate evidence [51]. *CrPIP2;3* is expressed in different tissues or organs at different expression levels in *C. rosea* plants (Appendix A). We characterized some special *cis*-acting elements in the *CrPIP2;3* promoter region (Figure 2), which indicated that the expression of *CrPIP2;3* is controlled by specific transcription factors that interact with these elements to induce the expression of this gene in response to some abiotic stresses or specific developmental signals in *C. rosea*. It is still unclear if *CrPIP2;3* is involved in the adaptive evolution of *C. rosea* to the extreme environment on tropical reefs or the coast. In the present study, we mimicked some abiotic stresses, such as high salinity (600 mM NaCl), alkali toxicity (150 mM NaHCO_3_, pH8.2), high osmotic conditions (300 mM mannitol), and ABA treatment (100 μM). The expression pattern of *CrPIP2;3* varied with stress treatment duration, generally first decreasing and then increasing (Figure 3). Alkali toxicity caused a slight temporary increase in *CrPIP2;3* expression in the root (2 h), following which the expression decreased. This indicated that alkali stress could temporarily enhance root hydraulic conductivity in *C. rosea*, and with the increase in alkali stress treatment, the water balance could be maintained even though the expression of *CrPIP2;3* decreased in the root. This could be because the water imbalance caused by alkali toxicity was only slight. High osmotic stress strongly induced *CrPIP2;3* expression (about 25 times) in *C. rosea* roots after 48 h, which may suggest that the high expression of *CrPIP2;3* could help to improve water absorption from the outside by the roots, and the higher expression of *CrPIP2;3* in the vine could help coordinate water transport to the aerial parts of *C. rosea*. In the leaf, the expression of *CrPIP2;3* was obviously decreased under all challenges or ABA treatment, which might be a protective strategy to reduce water loss via the leaves by transpiration. In general, our *CrPIP2;3* expression results indicated that *CrPIP2;3* may participate in osmotic and salt-alkali tolerance in *C. rosea* and is therefore probably involved in the adaptation of *C. rosea* to tropical reefs or coasts.

As a unicellular eukaryote with a rapid growth rate, yeast has become a convenient protein expression system for protein functional identification. We found that inducing the expression of *CrPIP2;3* in yeast (Appendix A) resulted in obvious sensitivity to high salt and high osmotic stress (Figure 4), which is, to some extent, opposite to that observed in other plant PIP members, such as barley PIP2;5 (*HvPIP2;5*), foxtail millet *SiPIP3;1* and *SiSIP1;1*, and date palm *PdPIP1;2*, which improved salt and osmotic stress tolerance in yeast [49,52,53]. Similarly, mammalian *AQP1* and *AQP5* caused moderate growth inhibition under salt and hyperosmotic stress when expressed in yeast [54], and many AQPs have been confirmed as membrane transporters of H_2_O_2_ in the plant response to stress [55]. In this study, the expression of *CrPIP2;3* caused sensitivity to high salt and osmotic stress, probably due to the more rapid water loss caused by the accumulation of CrPIP2;3 in the cell membrane. This offered further evidence that CrPIP2;3 is active in water transmembrane transport in vivo. Accordingly, the sensitivity to H_2_O_2_ was due to the stronger import penetration to H_2_O_2_ and more toxic effects on yeast cells caused by *CrPIP2;3* than the control yeast.

In order to elaborate the function of *CrPIP2;3* in abiotic stress in plants, we generated transgenic *A. thaliana* plants overexpressing the *GFP-CrPIP2;3* fusion gene under the control of the constitutive CaMV 35S promoter (Appendix A). In general, the accumulation of aquaporins in plants may alleviate water deprivation by improving water retention and maintaining water balance under unfavorable conditions, thereby maintaining plant survival even under serious water shortages or special circumstances. To date, it has been demonstrated in many plants that the heterologous expression of plant *AQP* genes can improve tolerance to salinity, drought, or oxidative stress [27,28]. In the present study, at the seed germination and seedling growth stages, *CrPIP2;3* overexpression generally only showed moderate improved tolerance to salt, salt–alkali, and high osmotic stress. This finding could be due to the ability of CrPIP2;3 to facilitate water absorption, being useful but not that marked when plants are grown on MS plates. The transgenic *A. thaliana* plants grown under NaCl, NaHCO_3_, or mannitol stresses showed stronger growth vigor than WT in terms of seed germination, hypocotyl elongation, and root length (Figure 5 and Figure 6). These results may suggest that the overexpression of *CrPIP2;3* could facilitate radicle or young root growth by maintaining the water management system under high osmotic or alkali toxicity. We also assessed the response of soil-cultivated adult plants to high salinity, drought, and alkali toxicity. Although 200 mM NaCl and 100 mM NaHCO_3_ did not cause any phenotypic differences between the WT and *CrPIP2;3 OX* lines, the drought-stressed *CrPIP2;3 OX* plants were able to recover after re-watering and showed a much higher survival rate than WT (Figure 7). This indicated that CrPIP2;3 could increase the water usage under drought, whereas high salt or alkali toxicity were not related to water deficit overmuch, but rather to an ion or pH imbalance. This result provided further evidence that *CrPIP2;3* confers higher water uptake in the roots at the adult plant stage.

According to the ROS levels determined by NBT and DAB staining in the leaves of *CrPIP2;3 OX A. thaliana* lines, our results showed that although salt, alkali, and high osmotic stress all resulted in ROS production in the plants, the effects of ROS accumulation caused by *CrPIP2;3* overexpression seemed to be negligible (Appendix A). Combined with our qRT-PCR results of *A. thaliana* antioxidant-enzyme genes (*CAT1*, *CSD1*, *APX1*, and *FSD1*), the overexpression of *CrPIP2;3* in *A. thaliana* did not affect the expression of these genes in the *CrPIP2;3 OX* seedlings. These results implied that the ectopic expression of *CrPIP2;3* in *A. thaliana* did not result in significant differences in the redox state in the *CrPIP2;3 OX* plants under abiotic stress. Furthermore, the phenotypic difference between the WT and *CrPIP2;3 OX* lines to stress tolerance was mainly caused by water absorption, not by antioxidant ability. Even though CrPIP2;3 is an active H_2_O_2_ transporter (Figure 4D,E), it seemed that, at least in response to water-deficit stress, CrPIP2;3 played a greater role in the transport of water rather than in ROS signaling.

In conclusion, the first aquaporin gene *CrPIP2;3* from the halophyte *C. rosea* was isolated and characterized systematically in this study and was strongly induced under salt, high osmotic stress, and ABA treatment. We demonstrated that the induced expression of *CrPIP2;3* in yeast could result in the sensitivity of yeast cells to salt, high osmotic stress, and H_2_O_2_, which indicated that CrPIP2;3 is an active H_2_O and H_2_O_2_ transmembrane transporter. Further, the overexpression of *CrPIP2;3* in *A. thaliana* conferred tolerance to salinity, drought, and salt-alkali in plants by retaining a better water status rather than by reducing ROS accumulation and membrane damage by enhancing the expression of antioxidant enzymes. This is the first study aimed at functionally characterizing an aquaporin gene from *C. rosea*. This study also represents the first attempt to elucidate the function of *CrPIP2;3* in plant genetic improvement under salt or drought tolerance. However, further research is required to better understand the function of *CrPIP2;3* in *C. rosea*, including its involvement in maintaining good cellular water status or interacting with other protective proteins, and thus participating in the adaptation of *C. rosea* to the extreme environmental habitats of tropical coasts and reefs.

## 4. Materials and Methods

### 4.1. Plant Materials, Growth Conditions, and Treatments

The *C. rosea* seeds were gathered from Yongxing Island (YX, 16°50′ N, 112°20′ E). The seeds were dried outdoors in summer, and the seedlings were cultivated outdoors with nutrient vermiculite or sandy soil until flowering and seed maturation from April to December (2019) in the South China Botanical Garden (SCBG, 23°18′ N, 113°35′ E). Seedlings of *C. rosea* were used for stress treatment assays to detect the expression patterns of *CrPIP2;3*. Several stress factors and hormone treatments, including salt (600 mM NaCl), alkali (150 mM NaHCO_3_, pH 8.2), water-deficit or drought (300 mM mannitol), and abscisic acid (ABA) treatment (100 μM), were applied to the *C. rosea* seedlings by irrigation or spraying to challenge the plants and to induce transcriptional changes in *CrPIP2;3*. The aerial parts (leaves and vines) and roots of *C. rosea* at different timepoints (0, 2, 12, 24, and 48 h) were collected separately and then immediately placed into liquid nitrogen for future use. The different tissues or organs of the *C. rosea* seedlings or adult plants present at SCBG were also gathered for further expression assays.

The wild-type (Col-0) *A. thaliana* was grown in a growth chamber at 22 °C with a photoperiod of 16 h light/8 h darkness. Transgenic *A. thaliana* plants were generated using the floral dip method. The T1 generation plants were selected by Basta (30 μL of 13.5% Basta solution added to 100 mL Murashige and Skoog (MS) medium) on MS plates, and the seedlings were identified by polymerase chain reaction (PCR) and reverse transcription (RT)-PCR to verify the overexpression of *CrPIP2;3* in *A. thaliana*. Three T2 homozygous lines were selected for further phenotype identification or subcellular localization analysis. The house-keeping gene *AtActin2* (*At3g18780*) was used as an internal control.

### 4.2. Gene Isolation and Bioinformatics Analysis

A full-length cDNA library from *C. rosea* was constructed using the SMART^™^ cDNA Library Construction Kit (Clontech, Takara Bio USA), with the *Saccharomyces cerevisiae* expression vector pYES-DEST52 as the carrier of the cDNA library. Dozens of colonies were picked randomly and sequenced, and one of the cDNAs showed intact open reading frames (ORFs) and had high homology to other plant *PIP2* genes. Combining the genomic DNA sequence results of *C. rosea*, this cDNA sequence was named *CrPIP2;3*. The obtained sequences of this *PIP2* gene were submitted to NCBI (Accession No.: MT787666). The protein sequences of CrPIP2;3 and other homologous proteins were aligned using ClustalW, and a phylogenetic tree was generated using the MEGA6 program with the neighbor-joining method and 1000 bootstrap replicates. The sequence information of other plant PIPs in this study is listed in Appendix A. The three-dimensional structures and transmembrane regions or orientation prediction of CrPIP2;3 were predicted by the Phyre2 server (http://www.sbg.bio.ic.ac.uk/phyre2/html/page.cgi?id=index). The conserved domain of CrPIP2;3 was searched using the Pfam online program (http://pfam.xfam.org/). The subcellular localization of CrPIP2;3 was also predicted using the WoLF PSORT server (https://wolfpsort.hgc.jp/) and Plant-mPLoc (http://www.csbio.sjtu.edu.cn/bioinf/plant-multi/).

To predict the motifs of the *CrPIP2;3* promoter region, the ATG upstream sequence (2000 bp) of the *CrPIP2;3* coding region was selected from the *C. rosea* genomic DNA sequence data (unpublished) and analyzed with the online program PlantCARE (http://bioinformatics.psb.ugent.be/webtools/plantcare/html/).

### 4.3. Yeast Strains and Functional Characterization of CrPIP2 in Yeast

For a further functional bioassay, the recombinant plasmid CrPIP2;3-pYES-DEST52 (Appendix A) and pYES2 (as a negative control) were transformed into *Saccharomyces cerevisiae* wild-type (WT) strains (BY4741 and W303) and *skn7Δ* via a polyethylene glycol (PEG)-lithium acetate-based transformation protocol. The yeast wild-type (BY4741) strain and H_2_O_2_-sensitive mutant yeast strain *skn7Δ* (Y02900, BY4741; *MATa*; *ura3Δ0*; *leu2Δ0*; *his3Δ1*; *met15Δ0*; *YHR206w::kanMX4*) were obtained from Euroscarf (http://www.euroscarf.de/index.php?name=News). The yeast wild-type W303 was kindly provided by Zhou et al. [56].

The transformed single colony was inoculated in SDG-Ura medium (plus 2% galactose) overnight at 30 °C, diluted with fresh pre-warmed SDG-Ura medium, and then incubated with vigorous shaking for approximately 48 h at 30 °C to reach an optical density just higher than 1.0 at OD600. Then, the cells were serially diluted in 10-fold steps, and 2 μL aliquots of each were finally spotted onto SDG-Ura medium plates with or without NaCl, sorbitol, PEG8000, or H_2_O_2_. The test plates were incubated at 30 °C for 3 to 7 d.

### 4.4. Expression Analysis of CrPIP2;3 in C. rosea Plants

Total RNA was isolated from *C. rosea* and *A*. *thaliana* using HiPure Plant RNA Kits (Magen, Guangzhou, China), and the cDNA was synthesized using TransScript One-Step gDNA Removal and cDNA Synthesis SuperMix (TransGen Biotech, Beijing, China) according to the manufacturer’s instructions. Quantitative reverse transcription (qRT)-PCR was conducted using a LightCycler^®^ 480 Gene Scanning system (Roche, Switzerland) and TransStart Top Green qPCR SuperMix (TransGen Biotech, Beijing, China). Gene expression levels were normalized using the *C. rosea* reference gene *CrEF-α* as internal control. The primer pairs (CrPIP2;3RTF/CrPIP2;3RTR and CrEF-αRTF/CrEF-αRTR) used for qRT-PCR are listed in Appendix A.

### 4.5. Generation of Transgenic A. thaliana Plants

The ORF sequence of *CrPIP2;3* was PCR-amplified with the primer pair CrPIP2;3OXF/CrPIP2;3OXR (Appendix A). The PCR fragments were then inserted between the *Eco*RI and *Bam*HI sites of the pGEAD plasmid to generate a recombination vector with a 35S promoter-driven overexpression cassette and Basta-resistant gene (Appendix A). After sequencing confirmation, the pGEAD plasmid and recombination vector containing *CrPIP2;3* were transformed into *Agrobacterium tumefaciens* GV3101, and transgenic *A. thaliana* was generated using the floral dip method. Positive transgenic plants with *CrPIP2;3* overexpression (*CrPIP2;3 OX* lines) were confirmed by genomic PCR with the primer pair CrPIP2;3OXF/CrPIP2;3OXR and by qRT-PCR with the primer pair CrPIP2;3RTF/CrPIP2;3RTR (Appendix A). The reference gene for the qRT-PCR was *AtACT2* (*At3g18780*) in *A. thaliana* using the primer pair AtACT2RTF/AtACT2RTR (Appendix A). The pGEAD transgenic plants were identified by a Basta-resistant screening protocol. In brief, three T3 homozygous (*OX 3#*, *OX 6#*, *OX 10#*) transgenic seeds were germinated for further research.

### 4.6. Evaluation of the Stress Tolerance of Transgenic A. thaliana

To evaluate the functions of *CrPIP2;3* in abiotic stress tolerance in plants, particularly the response to high salinity or alkali, water deficit, or drought-stress challenges, the seed germination rate, seedling root length, and the phenotype of adult plants of three *CrPIP2;3 OX* homozygous lines (*OX 3*#, *OX 6*#, *OX 10*#) under salt/alkali or drought stresses were recorded and analyzed separately.

The seed germination rates of *OX 3*#, *OX 6*#, *OX 10*#, and WT were measured after 7 d of germination. In brief, sterilized seeds were spotted on MS plates containing 150 mM, 175 mM, and 200 mM NaCl (salt stress), or 5 mM NaHCO_3_ plus 95 mM NaCl (pH 8.2) and 10 mM NaHCO_3_ plus 90 mM NaCl (pH 8.2) (alkali stress), or 200 mM, 300 mM, and 400 mM mannitol (hyperosmotic stress). The germination rates were counted (*n* = 30–50), and photographs were taken.

To evaluate the seedling tolerance to salt, alkali, or hyperosmotic stress, the root lengths of the *OX 3*#, *OX 6*#, *OX 10*#, and WT seedlings were also measured after 7 d of different stress challenges. In short, seeds were germinated on MS plates for 3 d under suitable conditions, following which the sprouts were transferred onto MS plates supplied with 100, 150, and 200 mM NaCl (salt stress), 0.5 mM NaHCO_3_ plus 99.5 mM NaCl (pH 8.2), 0.75 mM NaHCO_3_ plus 99.25 mM NaCl (pH 8.2), and 1 mM NaHCO_3_ plus 99 mM NaCl (pH 8.2) (alkali stress), or 200, 300, and 400 mM mannitol (hyperosmotic stress). The seedling lengths were calculated, and photographs were taken.

Salt, alkali, and drought tolerance assays were also assessed using transgenic adult *A. thaliana* plants. The *A. thaliana* seeds (*OX 3#*, *OX 6#*, *OX 10#*, and WT) were sown in vermiculite directly and were well-cultivated for 30 d. Subsequently, these adult plants were subjected to the following stress tolerance assays. For the salt tolerance assays, the plants were well irrigated with 200 mM NaCl solution for 21 d. For the alkali tolerance assays, the plants were watered with 100 mM NaHCO_3_ (pH 8.2) solution for 35 d. For the drought tolerance assays, the plants were maintained under continuous drought for 13 d and were re-watered for another 7 d. The survival rates were counted, and photographs were taken.

The reactive oxygen species (ROS) accumulation was also assessed with a nitro-blue tetrazolium (NBT) or 3.3′-diaminobenzidine (DAB) staining assay. In brief, three-week-old seedlings (WT and *CrPIP2;3 OX* lines) growing in the soil were soaked in 200 mM NaCl, 400 mM mannitol, or 50 mM NaHCO_3_ solutions for 24 h, with water as the control. The rosette leaves were collected before and after each treatment, and the in situ detection of H_2_O_2_ and O_2_^−^ in the leaves was determined with 1 mg/mL NBT or 1 mg/mL DAB solution, respectively, for 12 h, followed by clearing in 96% ethanol.

### 4.7. Expression Analysis of Stress-Responsive Marker Genes in CrPIP2;3-Overexpression A. thaliana

For detecting the expression of antioxidative system-related genes (*CAT1*, *CSD1*, *APX1*, and *FSD1*) and abiotic stress-related genes (*HAI2*, *RD26*, *RD29B*, and *ANAC19*) in three-week-old *A. thaliana* (WT or transgenic lines *OX 3#*, *OX 6#*, and *OX 10#*) planted in soil, the total RNA was isolated from the rosette leaves at different time points (with or without stress treatments, including 200 mM NaCl, 400 mM mannitol, or 50 mM NaHCO_3_ at pH 8.2), and cDNA synthesis was performed using the above procedure. The reference gene for the qRT-PCR was *AtACT2* (*At3g18780*) in *A. thaliana*. The primers used for qRT-PCR are listed in Appendix A.

### 4.8. Statistical Analysis

All of the above experiments were repeated at least three times independently, and the data shown are the mean ± SD. In this research, statistical analyses were performed using the statistical tools (Student’s *t*-test) of Excel 2010 software (Microsoft Corp., Albuquerque, NM, USA). The significance level was defined as * (*P* < 0.05) and ** (*P* < 0.01).

## Figures and Tables

**Figure 1 ijms-22-00565-f001:**
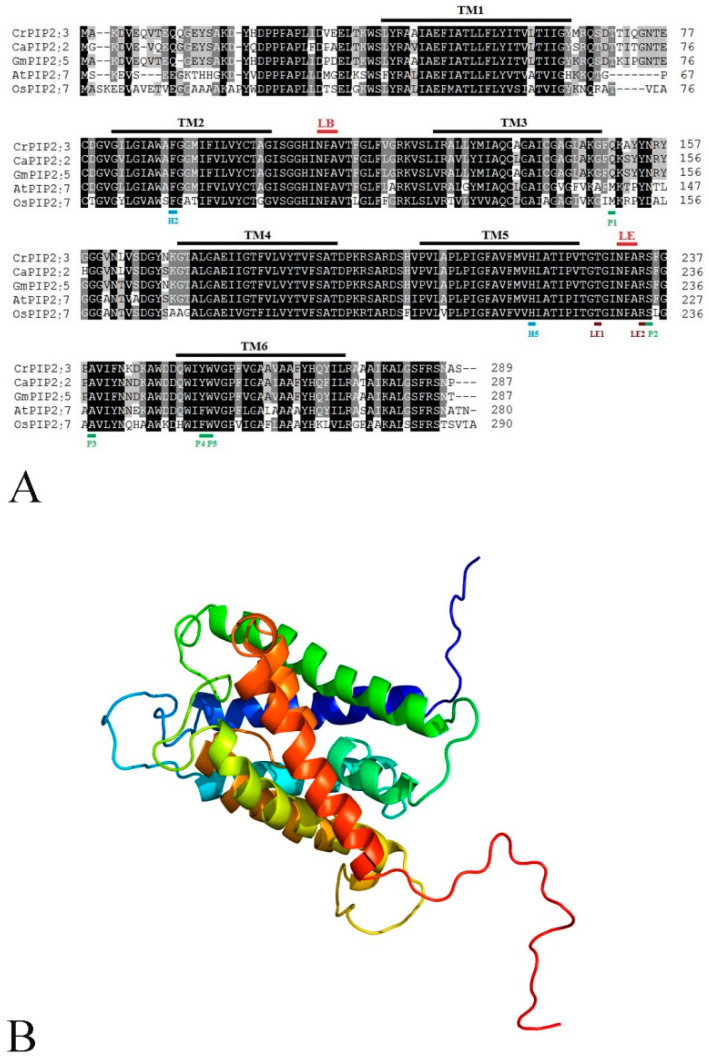
Protein sequence, transmembrane domain prediction, and phylogenetic analyses of the CrPIP2;3 protein. (**A**) Sequence alignment analysis of CrPIP2;3 with other known PIP proteins. The conserved amino acid residues in all proteins are highlighted in black and gray. The MIP domain (PF00230) is marked with a green frame. Six transmembrane helices domains (TM1–TM6) and two conserved Asn–Pro–Ala (“NPA”) motifs (LB and LE) are marked with black and red solid lines. The aromatic/arginine (ar/R) selectivity filters (H2, H5, LE1, and LE2) and Froger’s positions (FPs) (P1, P2, P3, P4, and P5) are marked with blue, brown, and green solid lines, respectively. The MIP signal sequence between TM2 and TM3 is also lined in a gray color. (**B**) Predicted 3D structure of CrPIP2;3 generated using the Phyre2 server. (**C**) The phylogenic tree of CrPIP2;3 and other PIP proteins from *A. thaliana* and soybean. The black triangle symbol showed the position of CrPIP2;3.

**Figure 2 ijms-22-00565-f002:**
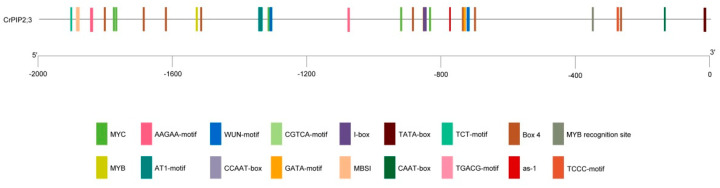
*Cis*-acting elements analysis of the putative promoter region of *CrPIP2;3* (ATG 2000 bp upstream).

**Figure 3 ijms-22-00565-f003:**
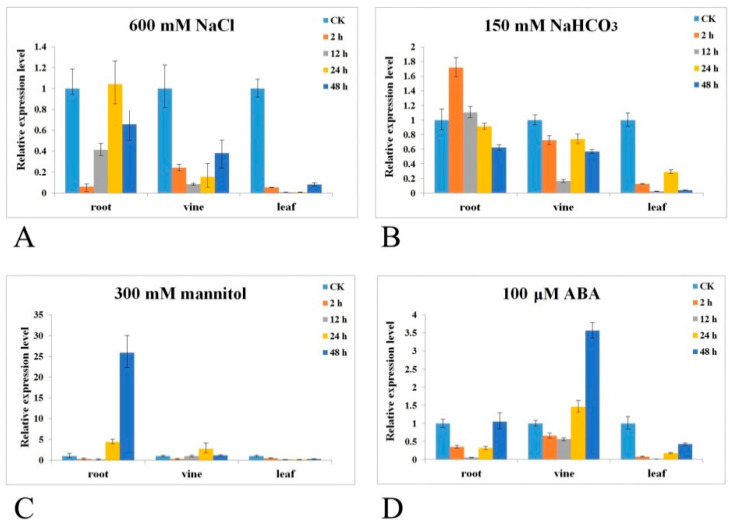
Expression pattern analyses of *CrPIP2;3* in *C. rosea* tissues. Time-course transcriptional levels of *CrPIP2;3* in the root, vine, and leaf in response to (**A**) salt; (**B**) salt–alkali; (**C**) high osmotic stress; and (**D**) ABA treatment. Error bars indicate the ±SD based on three replicates. In different tissues under various treatments, all of the expression levels were compared with CK (control check, the beginnings of each treatment).

**Figure 4 ijms-22-00565-f004:**
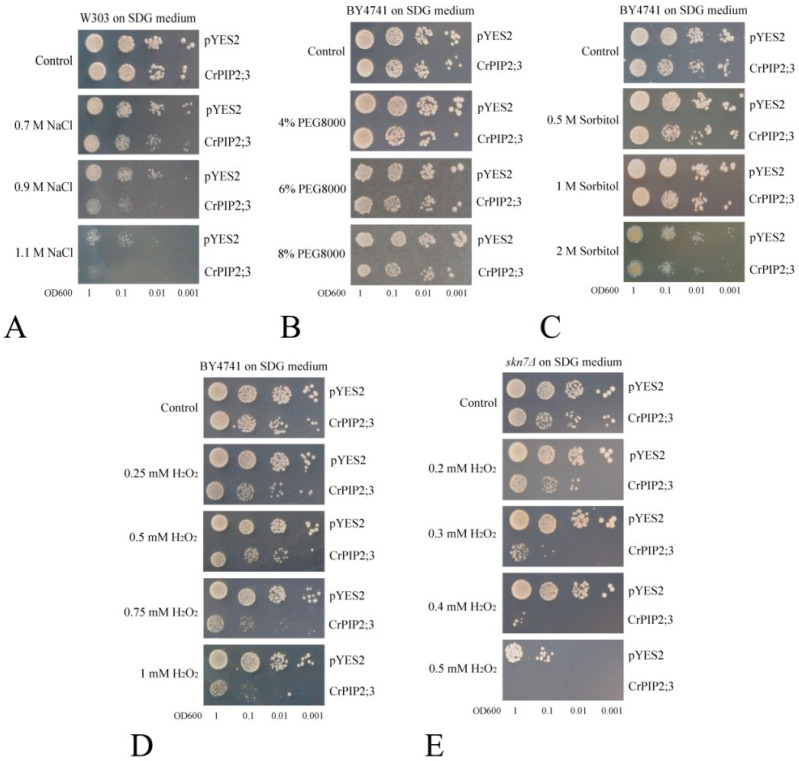
Spot assay for salt, osmotic stress, and H_2_O_2_ tolerance in yeast. Dot-spot assays for (**A**) salt-tolerance (different concentrations of NaCl) in the yeast wild-type (W303) expressing *CrPIP2;3* (carrying *CrPIP2;3*-pYES DEST52) or control (carrying pYES2); (**B**) high osmotic stress tolerance (different concentrations of PEG8000) in the yeast wild-type (BY4741); (**C**) high osmotic stress tolerance (different concentrations of sorbitol) in the yeast wild-type (BY4741); (**D**) oxidative stress tolerance (different concentrations of H_2_O_2_) in the yeast wild-type (BY4741), and (**E**) oxidative stress tolerance (different concentrations of H_2_O_2_) in the H_2_O_2_-sensitive yeast mutant strain *skn7Δ*.

**Figure 5 ijms-22-00565-f005:**
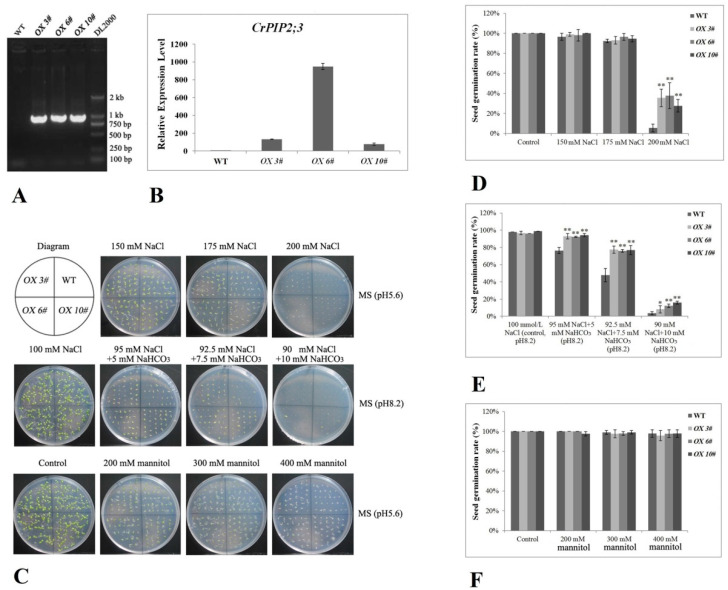
Overexpression analyses of *CrPIP2;3* in transgenic *A. thaliana* lines and stress analyses of transgenic plants with regards to the seed germination rate. (**A**) Genomic PCR analysis of *CrPIP2;3* in three transgenic *A. thaliana* lines (*CrPIP2;3 OX 3#*, *OX 6#*, and *OX 10#*) and WT plants; (**B**) quantitative RT-PCR analysis of *CrPIP2;3* in transgenic *A. thaliana* lines and WT plants; (**C**) photographs of transgenic lines and WT seeds germinated on MS medium alone or on MS medium with NaCl, NaCl plus NaHCO_3_ (pH 8.2), or mannitol for 7 d; (**D**–**F**) seed germination rates in WT and transgenic lines under NaCl (**D**), NaCl plus NaHCO_3_ (pH 8.2) (**E**), and mannitol (**F**) stresses after 7 d. Error bars indicate the SD based on at least three replicates (*n* ≥ 3). Asterisks indicate significant differences from the control (Student’ s *t*-test, * *P* < 0.05 and ** *P* < 0.01).

**Figure 6 ijms-22-00565-f006:**
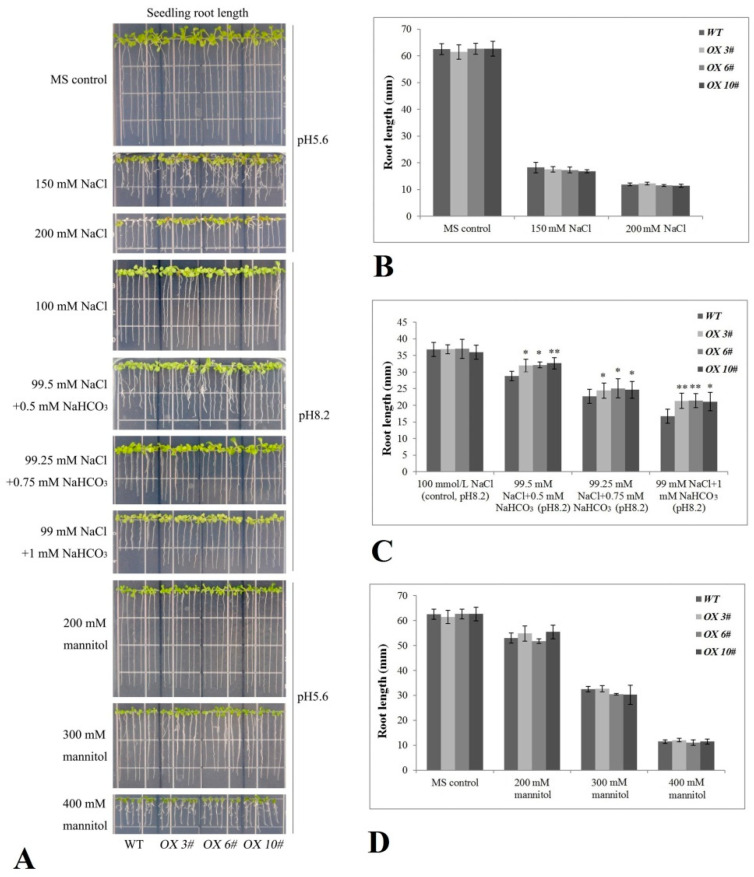
Salt, salt–alkali, and high osmotic stress analyses of transgenic plants with *CrPIP2;3* based on seedling root length. Four-day-old seedlings were transplanted into MS medium containing NaCl, NaCl plus NaHCO_3_ (pH 8.2) or mannitol and were then grown for 7 d before measuring the root length. (**A**) Photographs of transgenic lines (*CrPIP2;3 OX 3#*, *OX 6#*, and *OX 10#*) and WT seedlings on MS medium or MS medium with NaCl, NaCl plus NaHCO_3_ (pH8.2), or mannitol; (**B**–**D**) seedling root length (mm) of WT and transgenic lines under NaCl (**B**), NaCl plus NaHCO_3_ (pH 8.2) (**C**), or mannitol (**D**) stresses after 7 d. Error bars indicate the SD based on at least three replicates (*n* ≥ 3). Asterisks indicate significant differences from the control (Student’ s *t*-test, * *P* < 0.05 and ** *P* < 0.01).

**Figure 7 ijms-22-00565-f007:**
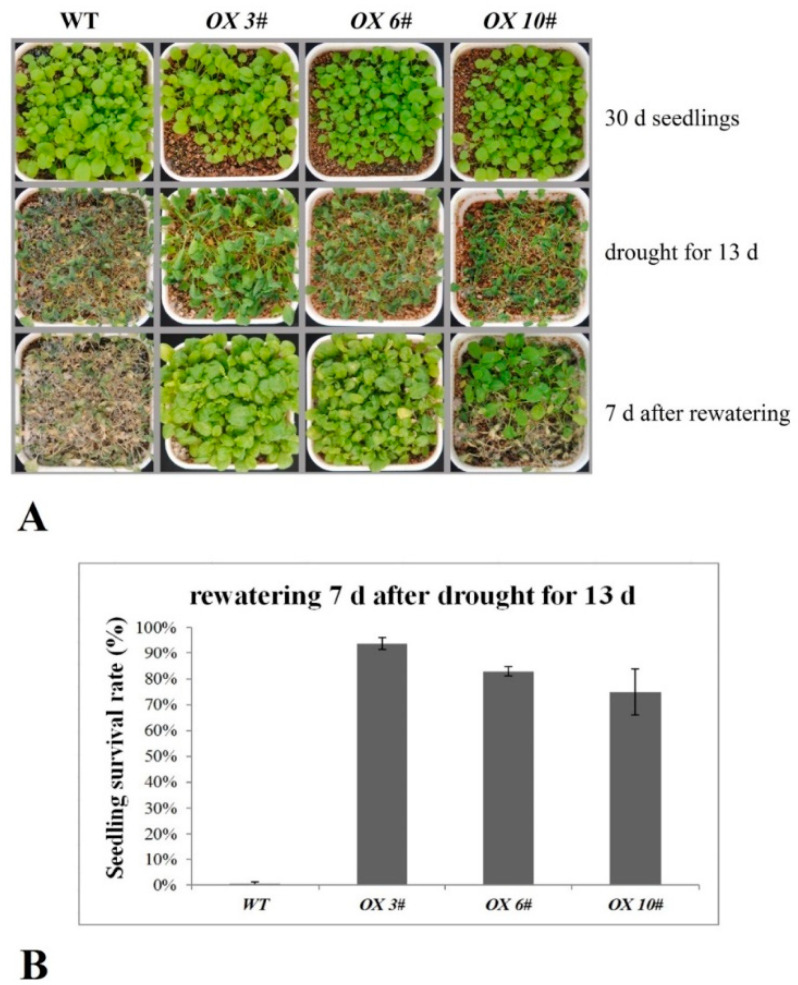
Photographs and survival rates of the *CrPIP2;3 OX* lines and WT plants grown in soil under normal and drought conditions. (**A**) The effects of withholding water on transgenic lines (*CrPIP2;3 OX 3#*, *OX 6#*, and *OX 10#*) and WT; (**B**) the statistics for the survival rate of the transgenic lines and WT *A. thaliana* after drought stress.

**Figure 8 ijms-22-00565-f008:**
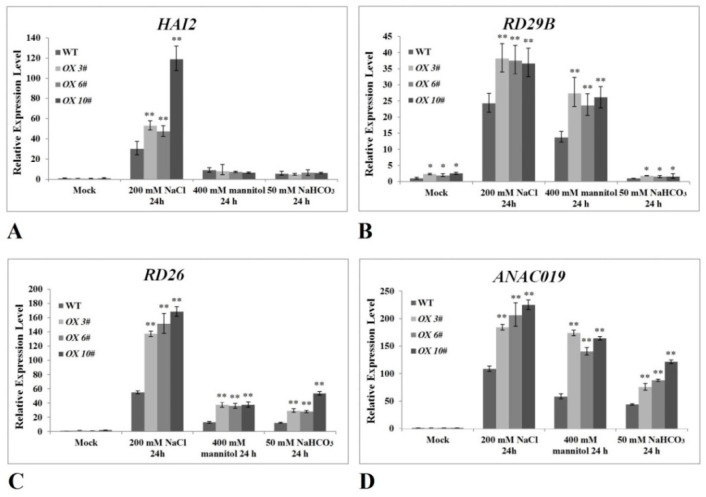
Analysis of the expression levels of stress-responsive genes in *CrPIP2;3* transgenic lines (*CrPIP2;3 OX 3#*, *OX 6#*, and *OX 10#*) and WT plants under normal and salt/alkali/osmotic conditions based on qRT-PCR. (**A**) *HAI2*; (**B**) *RD29B*; (**C**) *RD26;* and (**D**) *ANAC019*. Error bars indicate the ±SD based on three replicates. Asterisks indicate significant differences from the WT (control, Student’s *t* test, * *P* < 0.05 and ** *P* < 0.01).

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
