# Peer review of "Ectopic Expression of CrPIP2;3, a Plasma Membrane Intrinsic Protein Gene from the Halophyte Canavalia rosea, Enhances Drought and Salt-Alkali Stress Tolerance in Arabidopsis"

_ijms, 2021, doi:10.3390/ijms22020565_

Round 1

Reviewer 1 Report

Using aquaporin protein genes isolated from Canavalia rosea halophyte, the authors obtained transgenic Arabidopsis thaliana plants.

With a number of modern techniques, the authors investigated the response of the obtained transgenic plants to the action of abscisic acid and a number of stresses, in particular, salt, drought, the action of reactive oxygen species, and others. Transgenic plants showed the highest resistance to dehydration, compared to wild plants, which could be expected in advance.

The work is relevant and aimed at finding ways to increase the resistance of plants to a number of stresses, the results are important for use in creating stress-resistant crops.

In my opinion, the results presented in the manuscript correspond to the topic of the journal and can be published.

Author Response

Response:

Special thanks for your good comments and affirmations for this manuscript. We have made some language revisions which were showed in the revised manuscript. We hope the revised manuscript will meet the publication criterion of ijms.

Reviewer 2 Report

Major comments:

  1. Fig. 3 - To study the effect of salt treatment, 600 mM was used. It is not clear why this concentration was used. Is it a common salt treatment for C. rosea? Reference?
  2. Fig. 3- As for the NaCl treatment, it not clear why 100 mM of ABA was chosen as suitable concentration. In Arabidopsis, common treatments are at the nano-micro M levels. Using 100 mM of ABA might change pH and likely to have other global effects that are not ABA-related.
  3. Fig. 5- The quality of the images of GFP-CrPIP2;3 signals is not sufficient to perform sub-cellular localization assay. As stain for plasma membrane, FM4-64 is recommended.

Minor comments:

  1. Fig. 3 – it is not clear what CK  stands for (control?).
  2. Fig. 1C / Fig. 2B – font too small to well read.
  3. Fig. 6D – (and throughout the text) use either nmol/L or nM but not both.

Author Response

Response:

Special thanks for your good comments and suggestions. The revision points were strictly listed according to the comments in more detail below.

Major comments:

  1. Fig. 3 - To study the effect of salt treatment, 600 mM was used. It is not clear why this concentration was used. Is it a common salt treatment for C. rosea? Reference?

Response:

Considering that C. rosea belongs to littoral halophyte, and this species used to suffer from seawater flowing during its whole growth phase, so the salt challenge should be more intensive than normal salt stress used for model plants, such as Arabidopsis with 200 mM NaCl. The average salt concentration of seawater is 3.5%, and most of the salinity is NaCl. This concentration approximately equals to 600 mM NaCl. Here in this study, we used 600 mM NaCl solution simulated the seawater.

  • Fig. 3- As for the NaCl treatment, it not clear why 100 mM of ABA was chosen as suitable concentration. In Arabidopsis, common treatments are at the nano-micro M levels. Using 100 mM of ABA might change pH and likely to have other global effects that are not ABA-related.

Response:

Sorry for this mistake in Figure 3. We described this ABA concentration in “Materials and methods” as 100 μM. We corrected it in the revised Figure 3.

  1. Fig. 5- The quality of the images of GFP-CrPIP2;3 signals is not sufficient to perform sub-cellular localization assay. As stain for plasma membrane, FM4-64 is recommended.

Response:

The PI staining was usually used to label the cell apoptosis, while it cannot cross over the living cell membrane, so in our research, the red fluorescence signal adhere to the membrane of PI-stained roots (without any plasmolysis) also showed the intact plasma membrane. Of course, the FM4-64 is better.

Minor comments:

  1. Fig. 3 – it is not clear what CK stands for (control?).
  2. Fig. 1C / Fig. 2B – font too small to well read.
  3. Fig. 6D – (and throughout the text) use either nmol/L or nM but not both.

Response:

  1. We have explained the CK as “the beginnings of each treatment” both in the text and in the figure legend part. We also corrected the ABA concentration with “100 μM”, replacing “100 mM”.
  2. We separated Figure 1 as two parts: Figure 1A/B and Figure 1C, to improve the picture. As to Figure 2, we replaced it with a more convenient and intuitive diagram, and removed the previous figure 2 to the “Supplementary materials” part (Fig. S1). We also put the pictures of supplementary materials in the new order.
  3. We have corrected the expression “mol/L” with “M” both in text and in all figures.

Round 2

Reviewer 2 Report

Fig. 5

The authors did not provide an explanation to the poor quality of the quality of the images of GFP-CrPIP2;3. It must be improved if considered for co-localization assay. 

PI indeed is commonly used to isolate cell that go through processes of cell death, however it is not a marker for plasma membrane rather than cell wall. The staining looks diffusive probably due to long staining time that allow intrernalization of the stain.  Staining with FM4-64 is reliable as PM marker when is not allowed to be internalized (fast staining and visualization).

Author Response

Response:

Special thanks for kindly reminders and suggestions. Plant aquaporins have been denominated mainly based on their distribution in different subcellular compartments, and PIPs were primarily located in the plasma membrane (PM). The PIP’s trafficking is essential to modulate PIP’s activity and function, which has been considered as an important regulation way to mediate the PM and endomembrane system delivery and thereby affecting the conformation of aquaporin monomers or interactors (doi: 10.1016/j.tplants.2012.12.003). Here based on the bioinformatic analysis of CrPIP2;3’s subcellular localization and phylogenetic analysis, we proposed that CrPIP2;3 was mainly located in PM, while the CrPIP2;3’s trafficking should be present even in the ectopic overexpression transgenic plants. Furthermore, during our observation the green fluorescence signals of GFP-CrPIP2;3 and pEGAD-GFP control plants, we found that the GFP signals of GFP-CrPIP2;3 fusion protein were much more weaker than the GFP signals of pEGAD-GFP control. Probably the extremely weak GFP signal might come from the conformational change caused by additional CrPIP2;3 in GFP-CrPIP2;3 transgenic Arabidopsis. Therefore, in the process of acquiring and processing of the fluorescence signal (which might need long analysis time), the seedling roots of Arabidopsis might be damaged by PI solution, or even weak PI signals were magnified. This process could in some cases produce diffusive images. In view of the above considerations, we deleted the “Subcellular localization of CrPIP2;3” both in result and in method parts, and we think this deletion will not affect the integrity of this manuscript. And also, we have made some language revisions which were showed in this revised manuscript. We hope that the revised version will meet the publication criterion of ijms.

Round 3

Reviewer 2 Report

The authors decided to remove the 'sub-cellular localization' section in order to avoid from presenting the problematic GFP-CrPIP2;3 images and the PI effect, and instead to base the PM localization on bioinformatic analysis. Although it would be most informative to confirm the PM localization, the lack of such data does not take from the main conclusions of the manuscript. Therefore, I find the proposed version suitable for publication.